# Differentiation of Spatial Units of Genus *Euthynnus* from the Eastern Atlantic and the Mediterranean Using Otolith Shape Analysis

Rubén Muñoz-Lechuga [1,2], Fambaye Ngom Sow [3], Diaha N'Guessan Constance [4], Davy Angueko [5], David Macías [6], Alexia Massa-Gallucci [7,8], Guelson Batista da Silva [9], Jorge M. S. Gonçalves [10] and Pedro G. Lino [1,*]

1   Portuguese Institute for the Ocean and Atmosphere (IPMA), Avenida 5 de Outubro s/n, 8700-305 Olhão, Portugal; rubenmunozlechuga@gmail.com
2   Department of Biology, Faculty of Marine and Environmental Sciences, University of Cádiz, 11510 Puerto Real, Cádiz, Spain
3   Oceanographic Research Center of Dakar Thiaroye—CRODT/ISRA, LNERV—Route du Front de Terre, Dakar BP 2241, Senegal; ngomfambaye2015@gmail.com
4   Center of Oceanology Research, 29 Rue des Pêcheurs—BP V-18, Abidjan 01, Côte d'Ivoire; diahaconstance@yahoo.fr
5   General Directorate of Fisheries and Aquaculture, Libreville BP 9498, Estuaire, Gabon; davyangueko83@gmail.com
6   Spanish Institute of Oceanography, Oceanographic Center of Málaga, Puerto Pesquero s/n, 29460 Fuengirola, Málaga, Spain; david.macias@ieo.csic.es
7   AquaBioTech Group Central Complex, Naggar Street, Targa Gap, MST 1761 Mosta, Malta; alexia.massagallucci@gmail.com
8   Blue EcoTech Ltd., 55 Gardenia Independence Street, ZBG 02666 Zebbug, Malta
9   Animal Science Department, University Federal Rural of Semiárido, Av. Francisco Mota, 572—Bairro Pres. Costa e Silva, Mossoró CEP 59.625-900, RN, Brazil; guelson@ufersa.edu.br
10   CCMAR—Centre of Marine Sciences, Campus de Gambelas, University of Algarve, 8005-139 Faro, Portugal; jgoncal@ualg.pt
*   Correspondence: plino@ipma.pt

**Abstract:** The shape of sagitta otoliths was used to compare individuals of little tunny (*Euthynnus alleteratus*) harvested on board commercial fishing vessels from the coastal areas along the Eastern Atlantic, including the Mediterranean Sea. Fish sampling and selection was designed to cover possible seasonal changes and tuna size. The research encompassed both morphometric and shape analyses of left sagittal otoliths extracted of 504 fish specimens. Four shape indices (Circularity, Roundness, Rectangularity, and Form-Factor) were significantly different between two groups, showing a statistical differentiation between two clear spatial units. The degree of divergence was even more pronounced along the rostrum, postrostrum, and excisura of the generated otolith outlines between these two groups. One group corresponds to the samples from the coastal areas in the Northeast Temperate Atlantic and Mediterranean Sea (NETAM Area) and a second group from the coastal areas off the Eastern Tropical Atlantic coast of Africa (ETA Area). This study is the first to use otolith shape to differentiate tunas from separate spatial units. These results could be used to re-classify previously collected samples and to correct time series of data collected.

**Keywords:** stock delimitation; tuna; morphology; fish population; shaper

**Key Contribution:** The analysis of the shape of sagitta otoliths of *Euthynnus alleteratus* statistically differentiated two spatial groups within the Eastern Atlantic. The results have implications for the spatial management of this species by ICCAT and could be used to separate historical samples.

## 1. Introduction

An accurate understanding of the fish population structure is of vital importance for sustainable management [1]. The incomprehension of the population structure of exploited species can lead to dramatic changes in biological attributes, productivity rates, and genetic diversity, as well as overfishing and depletion of the less productive population units [2,3]. To initially investigate the population structure of marine fish species, it is necessary to successfully determine the discrimination of the populations [4]. Several techniques have been used to identify stock limits, such as tagging experiments [5], analyses of spatial and temporal variation of genetic or morphometric markers [6–8], differentiation of life-history variables [9], parasites composition [10,11], and contaminant concentrations [12].

Otolith-based research is an important tool that provides information on the population biology and life history of fish that is otherwise extremely difficult to collect [13]. Otolith research can be categorized into several distinct themes: (1) age and growth estimation and validation; (2) chemical composition; (3) historical comparisons using ancient otoliths; (4) otolith identification from intestinal contents of marine species in diet-based studies; (5) species identification, especially among cryptic species or in particular environments; and (6) shape analysis for stock structure and fisheries management [14–21].

Otolith morphology study can be a powerful tool for fish stock identification purposes, in particular for stocks that are likely to have spent a significant part of their lives in different environments and therefore may provide an indirect basis for potential stock separation [4,22,23]. Otolith shape analysis has recently gained interest among fisheries biologists, as it is a particularly powerful tool to investigate the stock discrimination in fish species, since several studies show evidence that it is controlled by both genetic and environmental factors [24–26], and thus highly variable between species and populations [17,27,28]. Although this technique has been widely used for discriminating diverse species, it has not been frequently applied in tuna and small tuna species [29–31].

The group of small tunas include a large number of species [32], mainly from the coastal areas [33]. The genus *Euthynnus* is one of the best-known species groups and is composed of three species: *Euthynnus lineatus*, *Euthynnus affinis*, and *Euthynnus alletteratus*. The little tunny (LTA)—*Euthynnus alletteratus*—is distributed throughout tropical and temperate zones of the Atlantic Ocean, including the Gulf of Mexico and the Mediterranean Sea. It is a commercially important tuna species in regional fishing communities due to the large volume of their catches [34], and is mainly exploited by gears such as gillnets, handline, traps, and purse seines [35].

The International Commission for the Conservation of Atlantic Tunas (ICCAT) is the organization responsible for the stock assessment and management of the populations of tunas and tuna-like species across the Atlantic Ocean and its adjacent seas [36]. This includes small tuna species such as *Euthynnus alletteratus*. According to the ICCAT official catch statistics, this species account for a significant proportion of the total small tuna species production, representing in the last years around 15% of the total catch [37]. Diverse biological information on this species is available focusing on age and growth [38–44], reproduction [45–51], and stock assessment [32,35]. However, concerning stock structure, there are some knowledge gaps with a lack of concise information on this species [43,44]. At present, there are no clear stock boundaries defined for some small tuna species such as little tunny in the Atlantic Ocean. Generally, five stocks unit areas are defined by ICCAT for data collection and management purposes: Mediterranean Sea, Southwest Atlantic, Southeast Atlantic, Northwest Atlantic, and Northeast Atlantic [35]. Therefore, the advance in the knowledge of the stocks structure of this species will improve the understanding and management of its fishery in the future.

In this research, the main focus was to evaluate population differences based on the shape of the sagittal otolith for little tunny captured along the coastal areas of Eastern Atlantic Ocean and Mediterranean Sea. In the same way, the usefulness of otoliths was validated as an applicable technique in small tunas, so as to generate correct management recommendations on these fisheries.

## 2. Materials and Methods

### 2.1. Tuna Sampling and Otoliths Collection

Little tunny samples were collected between 2017 and 2021 by observers on board commercial fishing vessels of gillnets, trawlers, handlines, traps, and purse seines on the Atlantic and Mediterranean, including Malta, Portugal, Spain, Senegal, Côte d'Ivoire, and Gabon coastal waters. Fish selection was based on the capture area during all potential months to cover possible seasonal changes and tuna size. Sagittal otoliths were extracted from 504 fishes at the laboratory, were cleaned with ultrapure water, allowed to dry, and stored in Eppendorfs. Straight fork length (SFL) was measured for each specimen to the nearest mm ranging from 219 mm to 987 mm (mean 407 mm; SD 135) (Table 1). The left sagitta was used for otolith morphometry and shape analysis.

**Table 1.** Summary of length samples (cm) of little tunny collected across various areas and countries.

| Area/Country | N | Mean | Mín | Máx | Fishery |
|---|---|---|---|---|---|
| **NETAM Area** | **164** | **50.0** | **21.9** | **103.0** | |
| Malta | 4 | 94.5 | 91.0 | 103.0 | Handline |
| Portugal | 143 | 45.4 | 21.9 | 69.6 | Trap; Gillnet |
| Spain | 17 | 82.3 | 22.0 | 98.7 | Trawler; Handline; Purse seine |
| **ETA Area** | **340** | **40.7** | **27.0** | **87.2** | |
| Côte d'Ivoire | 91 | 45.0 | 30.5 | 87.2 | Gillnet |
| Gabon | 62 | 32.8 | 27.0 | 38.0 | Gillnet; Purse seine |
| Senegal | 187 | 41.8 | 30.4 | 61.0 | Handline; Purse seine |

### 2.2. Otoliths Image Processing and Shape Analysis

For imaging, otoliths were photographed individually using a digital video camera mounted on a binocular microscope (Nikon SMZ1270, Tokyo, Japan) under reflected light and dark field. Each otolith was oriented with the sulcus side facing up and the rostrum pointing to the left. In some cases, to improve the image quality of the otoliths, the software ImageJ 1.53 t was used [52]. All images were stored in JPEG format with file sizes ranging from 107 to 790 kb. The images were imported into the "shapeR" package [52] for R [53], and were analyzed under the same threshold level (0.2) to generate the otolith morphometrics including length, width, area, and perimeter (Figure 1).

### 2.3. Preliminary Data Analysis

Preliminary analyses were carried out comparing otolith shapes collected in coastal waters of different countries. To examine and compare the variation in otolith outline of each country, the mean shape was plotted using the "shapeR" package [54]. Standardized wavelet coefficients represented the otolith shape. Standardization of the wavelet coefficient uses the straight fork length of the fish to remove the allometric effect of growth on the otolith, but is unable to correct any ontogenetic changes which an otolith may experience across sizes/ages [25]. This adjustment for allometric relationships with fish length is also implemented in the "shapeR" package [52]. The standardized wavelet coefficients were visually inspected for normality before further statistical analyses. Canonical Analysis of Principal Coordinates (CAP) was conducted using the "capscale" function of the "vegan" package [55] on standardized wavelet coefficients.

Hierarchical cluster analysis was performed to evaluate the similarity and diversity of country samples based on the averages of the CAP ordination and the mean of six otolith morphological indices (Figure 2A). For cluster analysis, Euclidean distance measures and Ward linkage were used. The six common shape indices calculated using the otolith morphometrics were: Circularity, Roundness, Rectangularity, Form Factor, Aspect Ratio, and Ellipticity (Table 2).

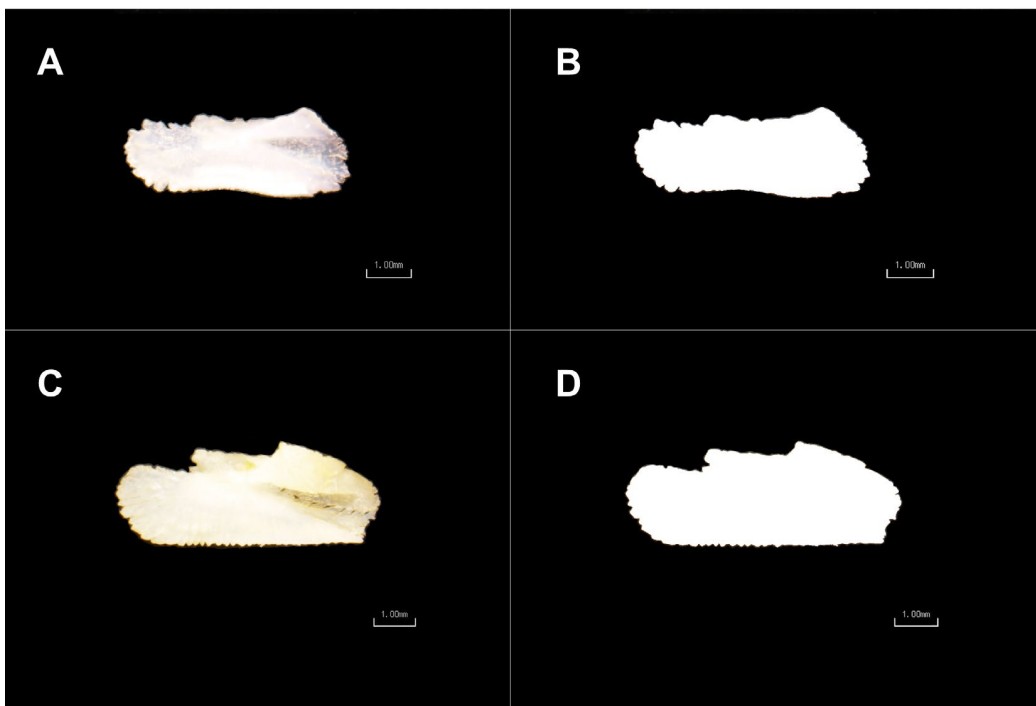

**Figure 1.** Original and processed images of two little tunny sagittal otolith from Portugal—58.9 cm SFL (**A**,**B**) and Côte d'Ivoire—67.9 cm SFL (**C**,**D**).

**Table 2.** Otolith morphological indices calculated from the measurement data. $O_A$ = otolith area (mm$^2$), $O_L$ = otolith length (mm), $O_P$ = otolith perimeter (mm), $O_W$ = otolith width (mm).

| Morphological Index | Formula |
|:---:|:---:|
| Aspect ratio | $\frac{O_L}{O_W}$ |
| Circularity | $\frac{O_P^2}{O_A}$ |
| Ellipticity | $\frac{O_L - O_W}{O_L + O_W}$ |
| Form-Factor | $\frac{4\pi O_A}{O_P^2}$ |
| Rectangularity | $\frac{O_A}{O_L * O_W}$ |
| Roundness | $\frac{4 O_A}{\pi O_L^2}$ |

The ordination of the averages in each country area was visually assessed using the first two canonical axes (CAP1 and CAP2) (Figure 2B). Finally, the average shape outline of otoliths for each country area was plotted to explore the relationship between otolith shape and country [56] (Figure 2C).

The preliminary analysis statistically separated two groups that corresponded to distinct spatial areas. The similarities obtained between individuals captured in nearby areas highlighted the clear differentiation between two groups. The first group included samples collected in the Northeast Temperate Atlantic and the Mediterranean Sea (Portugal, Spain, and Malta) called the NETAM Area (*n* = 164) and the second group included samples collected in the Eastern Tropical Atlantic coast of Africa (Senegal, Côte d'Ivoire, and Gabon), which was called the ETA Area (*n* = 340) (Table 1; Figure 3).

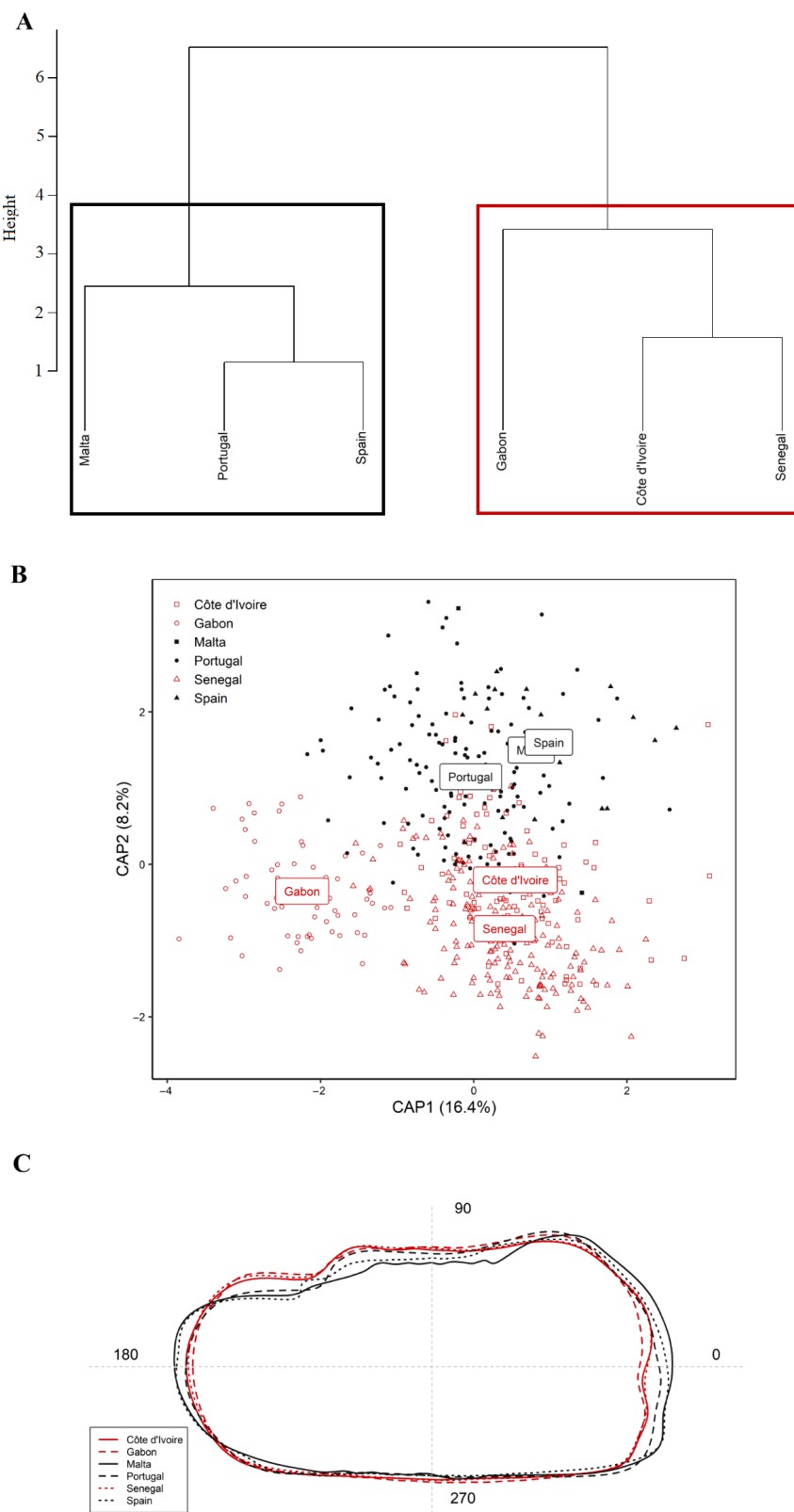

**Figure 2.** Hierarchical clustering based on the averages of CAP ordination and the otolith morphological indices (**A**). Canonical analysis of principal coordinates of the Wavelet coefficients for six country areas. CAP1 and CAP2 are the first and second discriminant axis, respectively. The group centroids represent the mean canonical value for each country samples analyzed (**B**). Average shape outline of otoliths for each country area. The numbers 0, 90, 180, and 270 represent angle in degrees (°) (**C**).

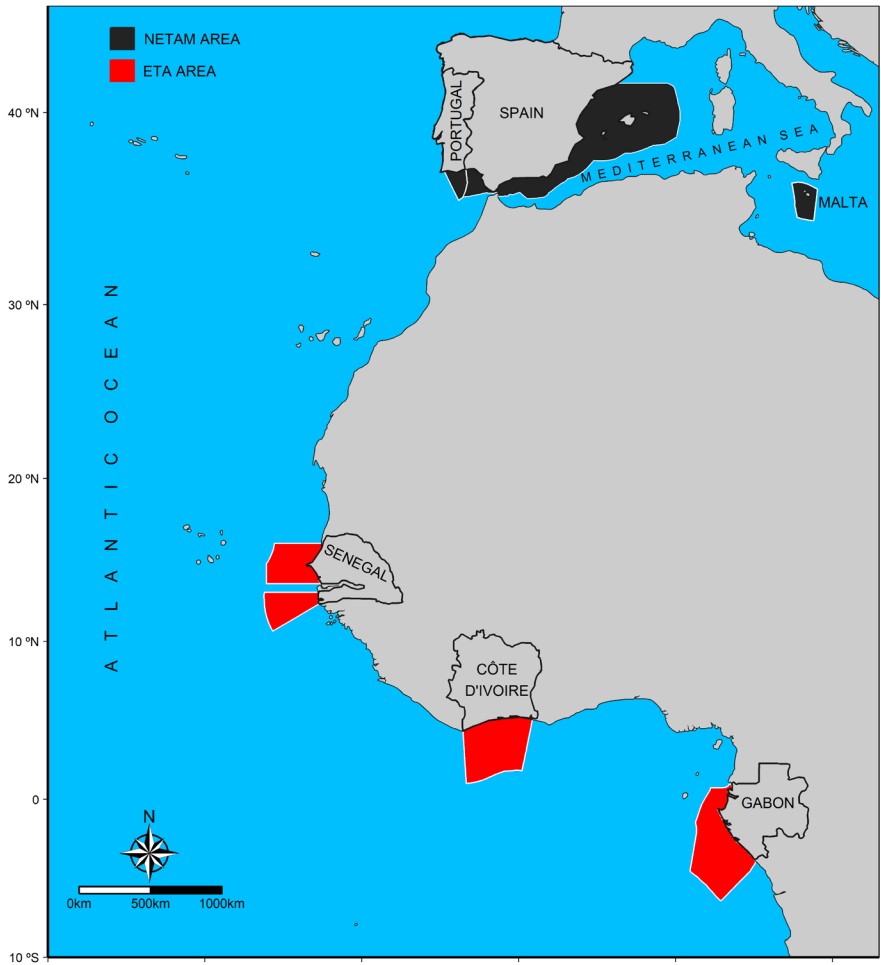

**Figure 3.** Map showing area where little tunny individuals were collected. The two distinct groups identified in the preliminary analysis (in black and red) correspond to samples collected in the Northeast Temperate Atlantic and Mediterranean Sea (NETAM Area) and in the Eastern Tropical Atlantic (ETA Area) respectively.

*2.4. Data Analysis*

In light of the existence of two statistically distinct groups, specific analyses were carried out to compare them. A univariate analysis of variance (ANOVA) was used to test for differences in the shape indices between individuals of two delimited areas (NETAM and ETA). Additionally, a *t*-test analysis was used to assess the differences in Wavelet coefficients between samples from both areas.

The variation in otolith shape was examined by plotting the mean shape of each area. To estimate which part of the otolith outline contributed most to the difference between the potential groups, the mean and standard deviation of the coefficients were plotted against the angle using "plotCI" function from the "gplots" package [57] as recommended by Libungan et al. (2016) [58]. The proportion of variation within groups along the outline was summarized with intraclass correlation (ICC). Canonical Analysis of Principal coordinates (CAP) was performed on the standardized wavelet coefficients to explore the relationships between otolith shape and geographical area. The ordination of the averages in each group area (NETAM and ETA) was graphically examined along the first two canonical axes (CAP1 and CAP2). The canonical scores were further tested for significance ($\alpha = 0.05$) using ANOVA-like permutation tests with 1000 permutations.

## 3. Results

The one-way ANOVA test used to compare shape indices between the delimited areas revealed differences (Table 3). In effect, examination of the mean otolith shape demonstrated that there were dissimilarities among both areas in the study. In this work, the differences were observed for four shape indices. The sagittae otoliths from the NETAM Area were significantly different from those from the ETA Area in Circularity, Roundness, Rectangularity, and Form-Factor ($p$-value $< 0.05$). In contrast, Aspect Ratio and Ellipticity were not significantly different. Among shape indices analyzed in this work, Circularity, Form-Factor, Roundness, and Rectangularity were the most efficient variables in distinguishing both delimited areas.

**Table 3.** One-way analyses of variance (ANOVA) to test differences in otolith shape indices between Northeast Temperate Atlantic together with the Mediterranean Sea (NETAM Area) and Eastern Tropical Atlantic (ETA Area) samples of *Euthynnus alletteratus*.

| Morphological Index | Mean ± NETAM | Mean ± ETA | F-Value | *p*-Value |
|---|---|---|---|---|
| Aspect ratio | 0.39 ± 0.04 | 0.39 ± 0.03 | 0.9 | 0.349 |
| Circularity | 30.86 ± 2.75 | 27.65 ± 2.28 | 190.9 | <0.001 |
| Ellipticity | 0.44 ± 0.04 | 0.44 ± 0.03 | 0.6 | 0.425 |
| Form-Factor | 0.41 ± 0.04 | 0.46 ± 0.04 | 187.2 | <0.001 |
| Rectangularity | 0.76 ± 0.03 | 0.80 ± 0.02 | 336.3 | <0.001 |
| Roundness | 2.50 ± 0.29 | 2.65 ± 0.20 | 44.6 | <0.001 |

The results of ANOVA-like permutation test using the Wavelet distances demonstrated significant differences between areas ($p < 0.05$). These differences were mainly at the excisura, rostrum, and postrostrum projections (Figure 4), which was further confirmed by examining variability in the mean Wavelet coefficients and the proportion of variation between both groups summarized with the ICC (Figure 5). Furthermore, almost 60% of the Wavelet coefficients revealed significant differences (Supplementary Figure S1). The otoliths from little tunnies captured in the ETA Area were less indented at the level of the excisura compared to those captured in the NETAM Area, but largest at the level of the postrostrum. This comparison of the otolith shape showed a large variation along the outline of the otolith at 0–40°, 120–210°, and 330–360° angles.

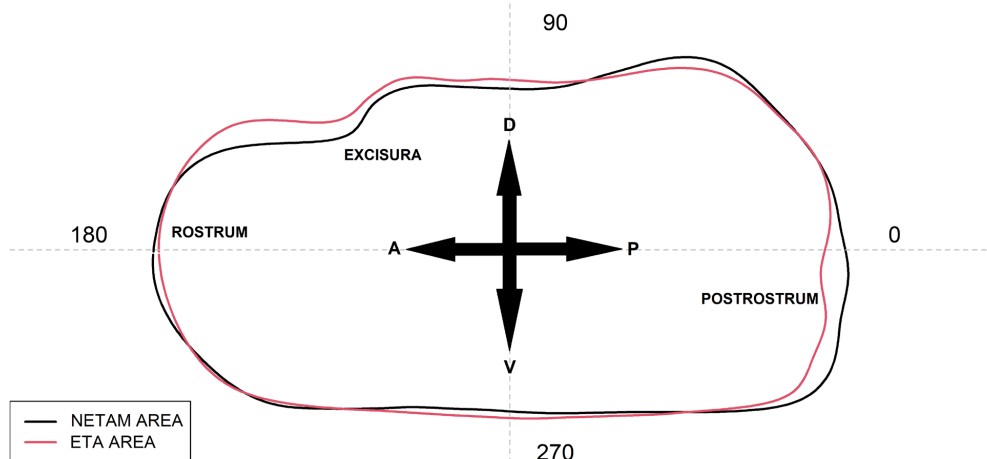

**Figure 4.** Average shape of otoliths for the two sampling areas in the study. The numbers 0, 90, 180, and 270 represent the angle in degrees (°) on the outline which correspond to Figure 5. Northeast Temperate Atlantic and Mediterranean Sea (NETAM Area). Eastern Tropical Atlantic (ETA Area). The central crossmap indicates the position of the otoliths: anterior (A); dorsal (D); posterior (P); ventral (V).

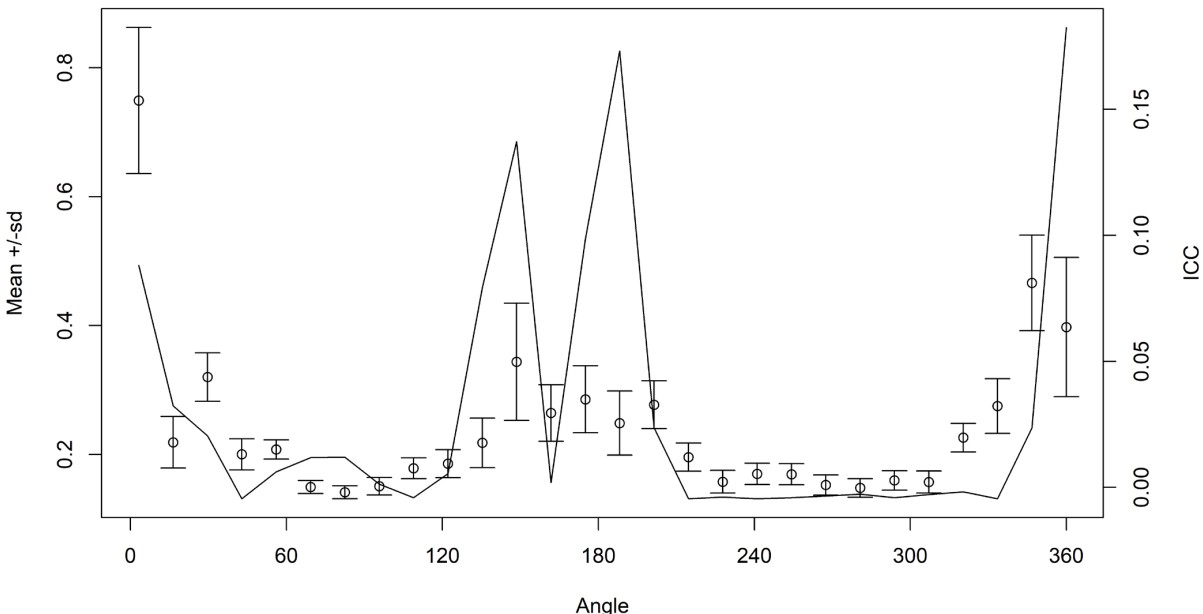

**Figure 5.** Mean and standard deviation (dots and whiskers) of the Wavelet coefficients for all combined otoliths and the proportion of variance within groups for the intraclass correlation (ICC, black solid line).

Analyzing the canonical scores for both areas revealed the largest differences between areas. The first two discriminating axes of the CAP analysis based on the Wavelet coefficients explained 36.6% of the variation between the two species group (CAP 1: 11.6%, CAP 2: 25%), demonstrating a clear difference between NETAM and ETA (Figure 6).

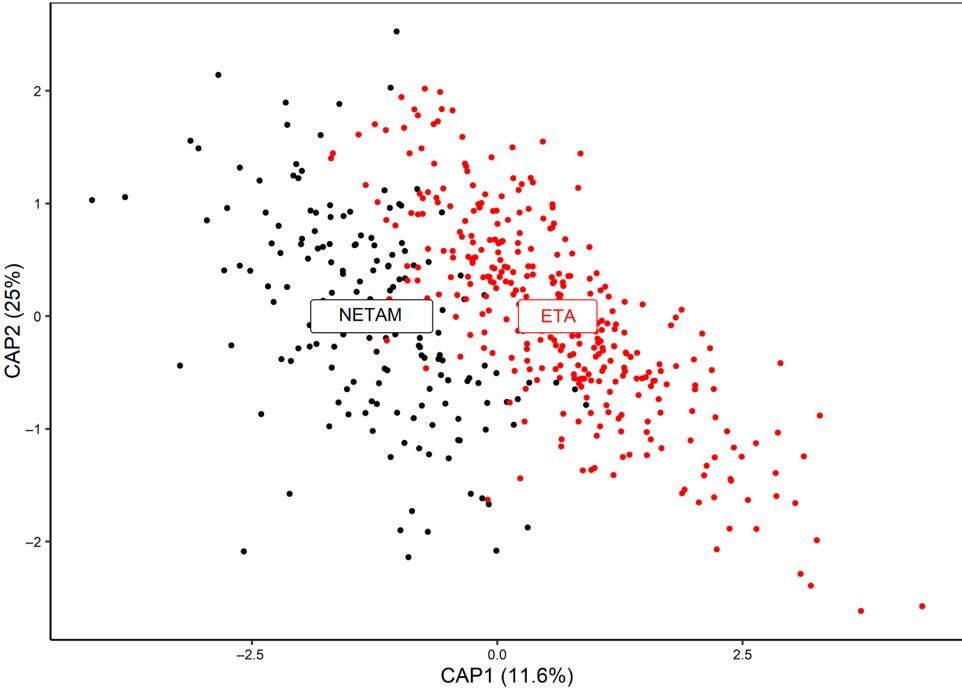

**Figure 6.** Canonical analysis of principal coordinates of the Wavelet coefficients for the two sampled areas. CAP1 and CAP2 are the first and second discriminant axis, respectively. The group centroids represent the mean canonical value for each area analyzed. Northeast Temperate Atlantic and Mediterranean Sea (NETAM Area). Eastern Tropical Atlantic (ETA Area).

## 4. Discussion

Otolith shape analysis statistical differences between the two groups provided a clear indication of the differentiation between these two spatial units of *Euthynnus alletteratus* present in the Atlantic Ocean. The most significant variation was observed in the shape-related morphometric indices, such as Circularity, Roundness, Rectangularity, and Form-Factor, which effectively differentiated the samples from NETAM and ETA areas (Table 3). The divergence was further evident along the rostrum, postrostrum, and excisura of the generated otolith outlines, as shown in Figure 4. These results were also supported by the proportion of variance within groups for the intraclass correlation (Figure 5).

This discreteness can be attributed to genetic isolation as well as the differences in environmental conditions between two delimited regions [59,60] creating phylogeographical breaks [61]. Another possible explanation for the differentiation of the species could be due to latitudinal isolation due to lack of migration patterns, geographic distance, and differences in oceanographic characteristics [62,63], which would also prove the pattern observed in the genus *Euthynnus* in the Atlantic [44]. This situation can lead to a biological specialization, which is linked to a differentiation in growth, reproductive, and morphological aspects [64].

Biological differences between both areas analyzed in this work have been poorly documented. In the genetic aspect, differentiating results could indicate the presence of separate species [65,66]. Olle et al. (2022) observed that the mtDNA CR divergence between the two areas studied of *E. alletteratus* was nearly 20 times larger than the CR divergence between *E. lineatus* (Pacific Ocean) and *E. affinis* (Indian Ocean), and similar to the distance that separates *E. alletteratus* respectively from *E. affinis* and from *E. lineatus* [65]. Regarding growth, almost all the works published so far come from very specific areas and mostly from the Mediterranean [39,40,42,43]. However, comparing one of the few studies in progress developed in the ETA area, specifically on the coast of Senegal [67], and comparing the length at age information with the values of other individuals in the Mediterranean Sea, differences are observed. For the same age, the values from Senegal presented smaller lengths compared to those from the Mediterranean and the Western Atlantic [41,44,68]. Similarly, reproduction has been poorly analyzed in this species. If the existing information between ETA and NETAM areas is compared, in a study carried out in the ETA area, a very extensive spawning period is observed, which occupies practically the whole year [47]. However, in the Mediterranean Sea, the period is quite seasonal and limited, coinciding with the warmer period, between the months of June and August [49–51,69]. However, both biological aspects must be studied in depth in the future. On the other hand, morphometric differences were observed between individuals from the Tunisian coast compared to others from the ETA area [70]. This evidence agrees with the spatial structure we propose in this study. Similarly, species differentiation along Atlantic Ocean have been documented for several fish genera such as *Lepidopus* spp. (Ward et al. 2008), *Auxis* spp. [71,72], *Thunnus* spp. [73], *Scomber* spp. [74,75], *Trachurus* spp. [76], *Zeus* spp. [77], and *Diplodus* spp. [78].

There is abundant scientific literature that applies otolith shape analysis as a stock differentiator or population structure descriptor [79–84]. This technique has also been widely used for the differentiation between species [85–91], demonstrating that it is an efficient technique for this type of analysis [28]. The shape of the otolith is known to vary depending on the ecological, evolutionary, and phylogenetic characteristics of each species [92]. This variation is particularly evident in coastal species that inhabit dynamic environments, such as *E. alletteratus*, and can be observed in the morphometry of their otoliths. The high differentiation and classification rates observed for our research-collected otoliths indicate that, as previously confirmed by genetic analyses, it follows that there are clear otolith shape differences between the Northeast Temperate Atlantic together with the Mediterranean Sea and Eastern Tropical Atlantic areas with a high degree of confidence. A pattern that is repeated in this type of study in the differentiation of species from shape otoliths, which is also fulfilled in our work, is that at least three otolith morphological descriptors analyzed

show significant variations between the groups of individuals analyzed and 25% of the otolith outlines present great divergences [85–91]. Perhaps these are results to consider when carrying out an analysis of this type for the differentiation between species. This may indicate certain isolation among localities that are nearby each other geographically or that there are natural environmental barriers that prevent mixing between both areas [93]. This situation is not common in pelagic fish species and even less in tuna, since they tend to migrate over medium or long distances, both along coastal areas and on open sea [94–97]. There are cases of restricted geographic expansion in some tuna species in the Atlantic waters, such as *Thunnus atlanticus*, which is distributed exclusively in tropical and subtropical waters of the Western Atlantic Ocean, ranging from the mid-Atlantic region of the United States east coast to northern Brazil, including the Gulf of Mexico [98]. In the same way, *Auxis thazard*, although it can be found throughout the Atlantic Ocean, is not distributed along the Mediterranean Sea, where *Auxis rochei* is dominant [71]. However, there are records mainly from the Strait of Gibraltar, the area where water masses interchanges between the Atlantic Ocean and the Mediterranean Sea [72].

From the analysis in the paper, the differentiation results are similar to other previously published results for different species of one same genus [27,91,99–101]. Some studies have been carried out using otolith shape analysis in tunas and small tunas, mainly for stock delimitation [29–31]. To our knowledge, however, this is the first published study to use otolith shape to validate tuna spatial units' differentiation that might correspond to different species. This could be applied to other genera with several species (e.g., *Thunnus* spp. or *Auxis* spp.) and would help to improve the accuracy of fisheries monitoring and facilitate re-classification of previously collected samples where the identification to the species level is problematic. Applying the technique to observer-sourced otolith collections would also improve confidence in datasets for analyses of the biological and ecological differences between species [102].

Otolith shape analysis complements the genetics study already published on stock structure of this species [65]. The findings from our research on little tunny in the Eastern Atlantic have unveiled a new perspective. There are clear indications of a species-level segregation between two distinct regions, which are presently regarded as a single species, challenging the existing geographic division at the stock level proposed by ICCAT [35]. This discovery holds significant implications for both the collection of scientific data and commercial fishing data, necessary for fisheries management. Furthermore, future investigations should focus on exploring new biological parameters, such as reproduction and growth, in both defined areas. A comparative analysis of these parameters could provide valuable insights into potential species-specific differences [103,104]. Moreover, to enhance our understanding of this phenomenon, additional samples are needed to gather comprehensive information from intermediate zones such as the waters of Morocco and Mauritania, as well as unexplored regions such as the Eastern Mediterranean and the Northeastern Atlantic.

## 5. Conclusions

This study demonstrates that the fish otolith shape can be utilized to validate the differentiation of tuna species. In addition to the limited literature available on the shape otoliths in tunas, no reports have been found regarding otolith asymmetries in these species. To address this issue, it is important to conduct a comprehensive comparative analysis of asymmetry in the left and right pairs of sagittal otoliths in the future. This analysis would greatly contribute to our understanding of this aspect and help fill this knowledge void. It is manifest that the pattern presented in this work does not comply with the current single accepted species *Euthynnus alletteratus* distributed along the Atlantic Ocean. A revised classification, considering the observed genetic and morphological evidence, should rather characterize the species of genus *Euthynnus* occupying the Atlantic Ocean, distinguishing Eastern Tropical Atlantic individuals as a new species of little tunny. To enhance the analysis in the future, it is crucial to include individuals from previously unexplored regions, such

as the Eastern Mediterranean or the Western Atlantic, as well as unanalyzed intermediate zones such as the coasts of Mauritania or Morocco. By expanding the geographical scope to encompass these regions, we could gain valuable insights into little tunny populations and their distribution in previously understudied areas. It is imperative to approach this issue rigorously, as it can lead to significant consequences for fisheries management.

**Supplementary Materials:** The following supporting information can be downloaded at: https://www.mdpi.com/article/10.3390/fishes8060317/s1, Figure S1: Median (white bar) and inter-quartile bounds (box) of Wavelet descriptors for the Northeast Temperate Atlantic together with the Mediterranean Sea (NETAM Area) and the Eastern Tropical Atlantic (ETA Area) of little tunny samples analyzed. Significant differences of *t*-test analysis between areas by descriptors are included in each graph.

**Author Contributions:** Conceptualization, R.M.-L. and P.G.L.; methodology, R.M.-L.; software, R.M.-L.; validation, G.B.d.S. and P.G.L.; formal analysis, R.M.-L.; investigation, R.M.-L.; resources, R.M.-L., F.N.S., D.N.C., D.A., D.M. and A.M.-G.; data curation, R.M.-L.; writing—original draft preparation, R.M.-L. and P.G.L.; writing—review and editing, F.N.S., D.N.C., D.A., D.M., A.M.-G., G.B.d.S. and J.M.S.G.; supervision, P.G.L. and J.M.S.G.; project administration, G.B.d.S. and P.G.L. All authors have read and agreed to the published version of the manuscript.

**Funding:** This research was funded by ICCAT Small Tunas Year Program (SMTYP) and partially by the European Union through the EU Grant Agreement No. S12.819116—Strengthening the scientific basis for decision-making in ICCAT.

**Institutional Review Board Statement:** Not applicable. All fish sampling involves deceased individuals obtained from the commercial fishing fleet, specifically purchased at the fish market. The Institutional Review Board approval is not applicable for this research.

**Data Availability Statement:** The raw data that support this study are available from the corresponding author upon reasonable request.

**Acknowledgments:** This work was carried out within the IPMA Portuguese National Program for Biological Sampling (PNAB), integrated in the EU Data Collection Framework (DCF). The authors are grateful to all people and members of the ICCAT Small Tuna Species Group who were of great help with the data and sampling collection. This work was carried out under the provision of the ICCAT Small Tunas Year Program (SMTYP). The contents of this paper do not necessarily reflect the point of view of ICCAT, which has no responsibility over them, and in no ways anticipate the Commission's future policy in this area. We express our gratitude to three anonymous referees whose valuable comments significantly enhanced the quality of our paper.

**Conflicts of Interest:** The authors declare no conflict of interest.

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
