# Peer review of "Differentiation of Spatial Units of Genus Euthynnus from the Eastern Atlantic and the Mediterranean Using Otolith Shape Analysis"

_fishes, doi:10.3390/fishes8060317_

Round 1
Reviewer 1 Report
Major comment:
The little tunny is an important stock in the coastal Atlantic Ocean, however its stock status has yet been well assessed. One of the reasons was lack of evidence of stock structure, which limits the data collection and statistics for fisheries capturing this species. Therefore, the result of this study is important to provide stock structure reference for stock assessment and management of ICCAT. In general , I recommend this manuscript be published after some minor revisions, as below:
Minor comments:
1, in the Abstract, there should be some key information regarding the data collection and sampling.
2, As this work is derived from the ICCAT research program, the main objective of the program is to provide information to facilitate stock assessment and management. Therefore, in the Introduction, I suggest the authors briefly introduced this piece of background. E.g., how was the stock assessment issues associated with this species around the eastern Atlantic.
3, In the Materials and Methods, I suggest to add a table listing the sample information (sample size, collection area/countries/, fishery, length ranges, sex if possible, etc.), to give a better understanding of how the samples are distributed.
4, I suggest to briefly discuss the implication of study's finding for the stock assessment and management, including the data collection improvement of the ICCAT associated with these kind of small tuna species.
Author Response
Responses to the reviewer 1 are included in the Word document.

Reviewer 2 Report
Despite I found this study innovative for tuna stock identification, it is affected by some methodological limitations and experimental design weaknesses in the present form. I suggest the authors evaluate and repeat some analyses to enhancing the value and the soundness of this study.
The introduction section needs to be enriched regarding otolith features, uses, and applications. Indeed, the topic appears in the manuscript in line 49 directly related to species/stock identification without introducing their biological and chemical features which gives this important role as a tool.
Line 89-90: "The left sagitta was used for otolith morphometry and shape analysis." why only the left? Could the authors motivate this choice with scientific support on regards? Indeed, several species show asymmetry on the sagittal otoliths, sometimes markedly. If not well documented and supported, this weakness could represent the main limitation of this study.
Moreover, the experimental design is affected by the analysis of all the samples together without a size class subdivision, which results fundamental in otolith studies as confirmed by several authors. Indeed, compare fish of different sizes and ontogenetic developmental stage could led to a bias in data validity and consequend discussion.
Best regards
The Reviewer
A moderate English language revision is needed to give more fluency to the manuscript.
Author Response
Responses to the reviewer 2 are included in the Word document.

Reviewer 3 Report
Muñoz-Lechuga et al. Differentiation of spatial units of Genus Euthynnus from Eastern Atlantic and Mediterranean using otolith shape analysis.
I read this article which deals with the shape of the sagittal otolith of the little tunny Euthynnus alletteratus to differentiate populations along the coastal areas of Eastern Atlantic Ocean and the Mediterranean Sea. The interest of such a study lies in the access to a better knowledge of the species, which should lead to the development of a better management policy for its stocks and fisheries. The findings of this study could also be applied to others genus with multiple species and would help improve the accuracy of fisheries monitoring and facilitate reclassification of previously collected samples where the identification to species level is problematic. The results of this study combined with strong genetic and morphological evidence may also lead to the revision of the current classification of the little tunny as the single species distributed along the Atlantic Ocean. The study covers an interesting question and is informative. In my opinion, this paper is well written with Introduction, Methods, Results, and Discussion well addressed. It deserves to be published after minor revision. I only have minor suggestions or comments as listed below.
L88-89 Authors stated :"Straight fork length (SFL) was measured for each specimen to the nearest mm ranging from 220 mm to 987 mm (mean 407 mm; SD 135)". I suggest that a summary statistic of the little tuna phenotype (SFL, weights, n, sex whether available) according to sampling sites L83-85, L134-140) be provided.
In L142-144 In table 1, there is an overlap between the numbering of the lines and the content of the table. Please improve it.
L159-160 "by [51]" = "by Libungan et al. [51]" I suggest this latter style to highlight the authors as well as easier reading.
L330-545 Please check abbreviation names of journal according to MDPI's guide for authors to improve their presentation in the reference list.
Author Response
Responses to the reviewer 3 are included in the Word document.

Round 2
Reviewer 2 Report
Dear Authors,
I have appreciated your care in revising the manuscript considering my previous comments, but I still have some comments in this regard.
Please consider within the period among lines 51-55 also these aspects:
- Species identification, especially among cryptic species or in particular environments such as transitional and deep-sea waters
10.3390/su14010398
10.1016/j.fishres.2020.105731
- Use of ancient otolith sampling for historical comparisons
10.1016/j.fishres.2023.106681
- Use of otolith from intestinal contents of marine species in diet-based studies
10.1007/s11160-021-09653-z
Moreover, about this comment:
Line 89-90: "The left sagitta was used for otolith morphometry and shape analysis." why only the left? Could the authors motivate this choice with scientific support on regards? Indeed, several species show asymmetry on the sagittal otoliths, sometimes markedly. If not well documented and supported, this weakness could represent the main limitation of this study.
The left otolith was selected for shape analysis, while the right otolith was chosen to examine growth through cross section analysis. After conducting preliminary analyses comparing several pairs (left - right), no discernible differences in shape or weight were observed for this particular species. Although limited literature on the topic exists, otolith asymmetries have not been reported in tunas. However, this aspect holds potential for further investigation in the future. Considering these factors, it was deemed to analyze both otoliths separately for various applications.
Why if you have conducted an investigation into the sagittal otolith pairs, you have chosen to not insert these important data within the manuscript? As you stated, data in these regards are still scarce and do not consider all the stocks of tunas, so would be essential to provide data in this regard in my opinion, this could seriously enhance the value of this manuscript and its interest for the researcher on this field, not only to support your experimental design.
Best regards
Author Response
Manuscript ID: fishes-2380086
Title: Differentiation of spatial units of Genus Euthynnus from Eastern Atlantic and Mediterranean using otolith shape analysis.
Journal: Fishes
Authors: Rubén Muñoz-Lechuga, Fambaye N. Sow, Diaha N'Guessan Constance, Davy Angueko, David Macías, Alexia Massa-Gallucci, Guelson Batista Da Silva, Jorge M.S. Gonçalves, Pedro G. Lino
Corresponding Author: Pedro G. Lino
-------------------
Key Contribution section has been included in the main document.
Changes of reviewer 2 are included in the main document and colored in blue.
Comments of Reviewers:
Reviewer 2
I have appreciated your care in revising the manuscript considering my previous comments, but I still have some comments in this regard.
Please consider within the period among lines 51-55 also these aspects:
- Species identification, especially among cryptic species or in particular environments such as transitional and deep-sea waters
10.3390/su14010398
10.1016/j.fishres.2020.105731
- Use of ancient otolith sampling for historical comparisons
10.1016/j.fishres.2023.106681
- Use of otolith from intestinal contents of marine species in diet-based studies
10.1007/s11160-021-09653-z
Thank you for the recommendation, we have improved the information about otolith research topics.
Moreover, about this comment:
Line 89-90: "The left sagitta was used for otolith morphometry and shape analysis." why only the left? Could the authors motivate this choice with scientific support on regards? Indeed, several species show asymmetry on the sagittal otoliths, sometimes markedly. If not well documented and supported, this weakness could represent the main limitation of this study.
The left otolith was selected for shape analysis, while the right otolith was chosen to examine growth through cross section analysis. After conducting preliminary analyses comparing several pairs (left - right), no discernible differences in shape or weight were observed for this particular species. Although limited literature on the topic exists, otolith asymmetries have not been reported in tunas. However, this aspect holds potential for further investigation in the future. Considering these factors, it was deemed to analyze both otoliths separately for various applications.
Why if you have conducted an investigation into the sagittal otolith pairs, you have chosen to not insert these important data within the manuscript? As you stated, data in these regards are still scarce and do not consider all the stocks of tunas, so would be essential to provide data in this regard in my opinion, this could seriously enhance the value of this manuscript and its interest for the researcher on this field, not only to support your experimental design.
Authors agree with the reviewer that this is an aspect that could be further addressed. Unfortunately, we do not have the information to carry out a statistical analysis comparing otolith pairs (left-right). A preliminary descriptive analysis was carried out and apparently no differences were appreciated, but it has not been possible to study this aspect in depth because the right otoliths have already been processed for age determination.
We intend to further investigate this aspect by acquiring new samples of individuals in the future.
Round 3
Reviewer 2 Report
Dear Author,
despite I still think about the strong limitation of the use of only an otholit and not pairs, the study now sounds more complete and reasonable.
Please highlight the limitations of this study and its future relapses in the conclusion section.
Best regards
Author Response
Despite I still think about the strong limitation of the use of only an otolith and not pairs, the study now sounds more complete and reasonable.
Please highlight the limitations of this study and its future relapses in the conclusion section.
The authors concur with the reviewer's assessment, about the limitations of this study regarding the asymmetrical approach and the absence of samples from unexplored regions, which have been duly included in the conclusion section.